# Efficacy of Treatment for Metastatic Hormone-Sensitive Prostate Cancer: An Umbrella Review of Systematic Reviews and Meta-Analyses

**DOI:** 10.3390/cancers15245714

**Published:** 2023-12-06

**Authors:** Pokket Sirisreetreerux, Napaphat Poprom, Pawin Numthavaj, Sasivimol Rattanasiri, Ammarin Thakkinstian

**Affiliations:** 1Department of Clinical Epidemiology and Biostatistics, Faculty of Medicine Ramathibodi Hospital, Mahidol University, Bangkok 10400, Thailand; pokket.sir@mahidol.edu (P.S.); sasivimol.rat@mahidol.edu (S.R.); ammarin.tha@mahidol.ac.th (A.T.); 2Department of Surgery, Division of Urology, Faculty of Medicine Ramathibodi Hospital, Mahidol University, Bangkok 10400, Thailand

**Keywords:** hormone-sensitive prostate cancer, metastatic prostate cancer, hormonal treatment, chemotherapy, umbrella review

## Abstract

**Simple Summary:**

Prostate cancer can be described as the second most common cancer worldwide in men. Systemic treatment initially with androgen deprivation therapy is the standard care for hormone-sensitive prostate cancer, aiming to reduce androgen receptors and result in tumor shrinkage. In addition, early combination treatment with androgen deprivation therapy (ADT) and novel anti-androgen agents or ADT with chemotherapy are recommended for selected patients. This umbrella review evaluated the medical treatment for hormone-sensitive prostate cancer, including ADT alone (bilateral orchiectomy, gonadotropin hormone-releasing hormone (GnRH) agonists and GnRH antagonists) or combination treatments (ADT with chemotherapy, ADT with enzalutamide, ADT with abiraterone and ADT with apalutamide). The results showed that combination treatments significantly improved OS compared with ADT alone in both OS and PFS outcomes with more adverse events, but no OS advantage of any combination regimen was observed over the others. However, high overlap of primary RCTs was found among meta-analyses.

**Abstract:**

Purpose: This umbrella review focused on evaluating the efficacy and adverse events of the metastatic hormone-sensitive prostate cancer patients receiving any treatment regimens, including ADT alone or combination treatments. Methods: This study conducted an umbrella review following the PRISMA 2020 checklist, aiming to summarize the available studies to evaluate the efficacy of medical treatments for metastatic hormone-sensitive prostate cancer. A literature search was performed to identify systematic reviews and meta-analyses (SRMAs) that included only randomized controlled trials (RCTs) up to September 2023. This study summarized their findings, evaluated overlapping data (i.e., the same RCTs were included in >one SRMA), tested for excessive significance (i.e., observed number of statistically significant studies > expected number by chance) and assessed the quality of the studies. Results: A total of 4191 studies were identified, but only 27 were included. Among those 27 studies, 12 were network meta-analyses and 15 were direct meta-analyses. Most studies showed no statistically significant difference in overall mortality among GnRH agonists, antagonists and bilateral orchiectomy. Combination treatment is more beneficial than ADT alone in both OS and PFS outcomes with more adverse events. Nevertheless, there is no OS advantage of any combination regimen over the others. Conclusion: Combination treatments demonstrated clear benefits in OS and PFS over ADT alone with more AEs. Further studies are needed to compare among combination treatments.

## 1. Introduction

Prostate cancer can be described as the second most common cancer worldwide in men, which is also the most frequently diagnosed cancer in more than 100 countries. It is estimated that there were almost 1.3 million new cases of prostate cancer and 359,000 associated deaths worldwide in 2018, which leads to it being the fifth most common cause of cancer death in men [1,2]. The mortality rate of prostate cancer was indicated to be around 6.7%, with geographical variations. Five-year survival varied among ethnicities, and it was highest among Asians (42.0%), followed by Hispanics (37.2%), American Indian/Alaska Natives (32.2%), black men (31.6%), and white men (29.1%). Although the prostate cancer screening program using PSA is recommended for a subgroup of the population, clinicians should provide information to those who would benefit from prostate cancer screening and make a decision based on a person’s values and preferences as per the American Urological Association’s guidelines, 2023. Currently, prediction algorithms including total and free PSA values are used to estimate the risk of high- or low-grade prostate cancer, which may be beneficial in aiding the decision-making process [3].

Metastatic disease can be manifested as bone pain, pathologic fractures, anemia, and paraneoplastic syndromes. Treatment of prostate cancer depends on the staging and risk group of the patients. Systemic treatment initially with androgen deprivation therapy (ADT) is the standard of care, aiming to reduce androgen receptors and result in tumor shrinkage, including bilateral orchiectomy, AR antagonism (i.e., inhibition of luteinizing hormone-releasing hormone (LHRH) and/or luteinizing hormone), and inhibition of androgen synthesis [4,5]. Additionally, the European Association of Urology (EAU)–European Association of Nuclear Medicine (EANM), the European Society for Radiotherapy & Oncology (ESTRO), the European Society of Urogenital Radiology (ESUR), and the International Society of Geriatric Oncology (SIOG) have suggested early combination treatment with ADT and novel anti-androgen agents or ADT with chemotherapy for metastatic hormone-sensitive prostate cancer (HSPC) in selected patients [6,7,8]. However, the results of the systematic reviews and meta-analyses (SRMAs) were still conflicting in some comparisons in terms of overall survival (OS), progression-free survival (PFS), and adverse events (AEs). In addition, the number of primary RCTs evaluating combination treatments were limited, and the overlap of the primary studies was a concern. Therefore, this study conducted an umbrella review that aimed to summarize the available SRMAs and to systematically appraise the results of the previous SRMAs, overlap and the quality of the studies.

## 2. Materials and Methods

This umbrella review focused on the treatment of metastatic HSPC patients receiving any treatment regimens, including ADT alone (bilateral orchiectomy, gonadotropin hormone-releasing hormone (GnRH) agonists, and GnRH antagonists) or combination treatments (ADT with chemotherapy, ADT with enzalutamide, ADT with abiraterone, and ADT with apalutamide). The umbrella review protocol was developed following the guidelines in the Preferred Reporting Items for Systematic reviews and Meta-analyses (PRISMA). The review protocol was registered with PROSPERO (CRD42020203546).

### 2.1. Search Strategy and Criteria for Study Inclusion

A literature search was performed for previous systematic reviews and meta-analyses using the following sources: MEDLINE via PubMed (www.ncbi.nlm.nih.gov/pubmed/ (accessed on 4 October 2020)), Scopus (www.scopus.com (accessed on 4 October 2020)), references of selected articles, Cochrane Central Register of Controlled Trials (CENTRAL), ClinicalTrials.gov, and conference proceedings. The search period was specified between 1 and 30 September 2020 and the search was updated every month by automatically setting it up in the PubMed and Scopus search engines. The last search was up to 30 September 2023. There was no language restriction. The search terms were constructed and followed the population, interventions and outcomes. The full search strategies are available in Appendix A.

Any systematic review with a meta-analysis (SRMA) was included if the study met the following criteria: SRMAs of randomized-controlled trials (RCTs), SRMAs that were conducted of patients with metastatic HSPC, SRMAs that reported the effect size of either ADT alone (bilateral orchiectomy, GnRH agonists and GnRH antagonists) or combination treatments (ADT with chemotherapy, ADT with enzalutamide, ADT with abiraterone, and ADT with apalutamide), and SRMAs that reported at least 1 outcome of OS, PFS and AEs. Studies with insufficient data after 3 attempts at contacting the author every 2 weeks and studies published in languages that reviewers could not translate were excluded.

### 2.2. Interventions and Outcomes

The interested interventions can be either ADT alone or a combination of ADT with other treatments. ADT alone consisted of bilateral orchiectomy, which can be total orchiectomy or subcapsular orchiectomy; a GnRH agonist, e.g., buserelin, goserelin, leuprolide and triptorelin; and a GnRH antagonist, e.g., degarelix, relugorix and abarelix. The combination treatment was defined as any kind of ADT given along with chemotherapy, e.g., docetaxel or enzalutamide or abiraterone or apalutamide. The comparators were selected as any specific medication within ADT alone, or bilateral orchiectomy in ADT alone, or any type of ADT alone, which could be bilateral orchiectomy or a specific medication or could be mixed methods of ADT for combination groups.

The primary outcomes included OS, PFS and AEs, and the secondary outcomes comprised quality of life (QOL) and the symptoms score. OS was defined as the time from randomization to death from any cause. PFS could be biochemical PFS (b-PFS), radiographic PFS (r-PFS), or clinical PFS (c-PFS), which was defined as the time from randomization to the first PSA rise or the occurrence of radiographic progression or disease progression, respectively. AEs were defined as adverse events of Common Terminology Criteria for Adverse Events (CTCAE) grade 3 or higher and serious AEs. Regarding secondary outcomes, QOL could be a general assessment of QOL or a disease-specific assessment of QOL.

### 2.3. Data Extraction and Study Quality Assessment

The search results were screened based on the title and abstract, and then data extraction was performed independently by 2 reviewers (PS, NP). Disagreements between the reviewers were discussed and solved by a team discussion and third reviewer (PN). The extracted data included characteristics of SRMAs, which included the last search date, number of included studies, type of intervention, pooled effect size with 95% confidence interval (CI), pooling methods (fixed-effect or random-effect), heterogeneity assessment, and publication bias. In addition, characteristics of individual RCTs were collected, including the total number of patients and number of events, effect size and *p*-value, and cited references of each RCT. The methodological quality/risk of bias of included SRMAs was assessed using ROBIS [9], which involved 3 phases as follows: phase 1 assessed relevance; phase 2 identified concerns with the systematic review process comprising 4 domains of study eligibility criteria, identification and selection of studies, data collection and study appraisal, and synthesis and findings; and phase 3 judged the overall risk of bias of the systematic review and concluded a low, high or unclear risk of bias.

### 2.4. Data Synthesis and Statistical Analyses in the Umbrella Review

Findings from each SRMA were described and reported. Data analysis for the umbrella review was performed concerning the degree of overlap of the overviews and the excess significance test [10]. For the degree of overlap, the covered area (CA) and corrected covered area (CCA) were reported. The CA was calculated by dividing the sum of the included publications by the product of the rows and columns. The CCA was calculated by dividing the frequency of repeated occurrences of the index publication in other reviews by the product of index publications and reviews using the equations below. A CCA score lower than 5 is considered a slight overlap, score of 6–10 is moderate overlap, score of 11–15 is high overlap and a value greater than 15 is considered very high overlap. CA and CCA were calculated using the following formula:CA = N/rc.
CCA = N−r/rc−r.
where N is the number of included publications, r is the number of rows, and c is the number of columns.

In addition, the exploratory test for an excess of significant findings of the pooled effect size from bias was performed by comparing the observed number of studies with positive results to the expected number of studies with positive results using the Chi-square test [11]. Excess significance was considered at a level of significance <0.10. All analyses were performed using STATA^®^ version 16.1 (StataCorp LP, College Station, TX, USA).

## 3. Results

A total of 4191 studies were identified, but only 27 SRMAs were included (see Figure 1). Among the 27 SRMAs, 12 were network meta-analyses and 15 were direct meta-analyses. ADT alone was compared in five studies and combination treatments were assessed in twenty-two studies. Regarding combination treatments, the OS of the patients receiving combination treatment was summarized in 19 studies and PFS was reported in 17 studies. However, no SRMAs of ADT alone reported OS, only the RR of overall mortality, and only one study described PSA-PFS in the patients receiving ADT alone. Twelve studies assessed the AEs of the treatments, five of ADT alone and eight of combination treatments. The study characteristics of the included systematic reviews and meta-analyses are shown in Table 1.

### 3.1. Mortality Rate

There were five SRMAs included in the study. Four studies reported overall mortality. One SRMA pooled five RCTs and compared the overall mortality between GnRH agonists and bilateral orchiectomy and found no statistical significance (RR 0.93, 95% CI 0.86, 1.00, I^2^ 0%) [12]. Among the five SRMAs that compared GnRH antagonists with GnRH agonists [12,13,14,15,16], the mortality rate was reported in three studies [12,13,16]. One SRMA showed a benefit of GnRH antagonists over GnRH agonists (RR 0.48, 95% CI 0.26, 0.9, *p* = 0.02, I^2^ 0%). However, two other SRMAs, pooled three RCTs, and one RCT reported no significant difference in overall mortality between the two medications (Figure 2).

### 3.2. Overall Survival

There was no OS reported for ADT alone, whereas 19 studies reported OS either among combination arms or combination arms compared with ADT alone. Regarding OS, docetaxel, abiraterone and apalutamide significantly improved OS compared with ADT alone [8,17,18,19,20,21,22,23,24,25,26,27,28,29,30,31,32]. However, Landre et al. assessed the effect on patients older than 70 years of age and found no statistically significant difference between docetaxel and abiraterone compared to ADT alone [31]. Regarding enzalutamide, two out of three SRMAs found a significant benefit in OS [25,26,29,30]. Most of the studies reported an indirect comparison among each pair of combination treatments. Only one RCT reported a direct comparison between abiraterone with ADT and docetaxel with ADT [26]. This study showed no differences in terms of OS. Most of the SRMAs showed no superiority of any combinations over the others in the OS outcome (Figure 3) [25,26,29,30].

### 3.3. Progression-Free Survival

Among ADT alone, only one SRMA mentioned b-PFS from one RCT and found no statistical significance for GnRH antagonists over GnRH agonists (RR 1.02, 95% CI 0.69, 1.50, *p* = 0.92, I^2^ 42%) [13]. Regarding PFS in the comparison between combination arms and ADT alone, all combination treatments showed a benefit over ADT alone [8,17,18,19,20,21,23,24,25,26,27,29,30,32]. Among combination treatments, an indirect comparison demonstrated no difference between enzalutamide and abiraterone [25,26,29,30] and between enzalutamide and apalutamide [25,26,29,30]. On the other hand, the comparisons between abiraterone and docetaxel [6,7,21,23,24,25,26,29,30,33], apalutamide and docetaxel [25,26,29,30] and apalutamide and abiraterone [25,26,29,30] were still conflicting. The forest plots of HRs are shown in Figure 4.

### 3.4. Adverse Events

For the AEs, an injection site skin reaction/irritation was significantly increased in the GnRH antagonist group. Two SRMAs found that GnRH antagonists had less cardiovascular events than GnRH agonists [13,16]. Acute MI [12,16], stroke, grade 3 or more AEs and severe life-threatening AEs were not statistically significantly different between two medications. Comparing abiraterone with ADT alone, abiraterone increased the risk of cardiovascular events [6,26], hepatic toxicity [6] and grade 3 or higher AEs, but lowered the risk of anemia [26]. However, severe, life-threatening AEs were only linked with ADT alone. Docetaxel significantly increased the risk of neutropenia, febrile neutropenia [20,26], and grade 3 or higher AEs, but they were not severe, life-threatening AEs. The forest plots of AEs are shown in Figure 5.

### 3.5. Excess Significance Test

Among the 19 SRMAs that reported OS, only 16 SRMAs had data available to evaluate the excess significance test. With statistical significance set to *p* < 0.1, no excess significance was shown from the analysis (Appendix A).

### 3.6. Degree of Overlap

Regarding the degree of overlap overall, the included cited number of RCTs was 127, with 27 duplicated RCTs. The degree of overlap was calculated and found to be 3.82% of the corrected covered area, which could be interpreted as a slight degree of overlap (Appendix A). On the other hand, when considering only studies that reported OS, the included cited number of RCTs was 33, with 17 duplicated RCTs. CCA was much higher at 16.4%, which indicated a very high degree of overlap. The citation matrixes are shown in Appendix A.

### 3.7. Risk of Bias Assessment

Among the 27 SRMAs, 17 studies (63%) were identified as having a low overall risk of bias and 10 SRMAs (37%) were identified as having a high risk of bias. The risk of bias assessment is shown in Table 2.

## 4. Discussion

Currently, various treatment options have been introduced for metastatic HSPC, including ADT alone, combination treatment with ADT and chemotherapy/novel anti-androgen/external beam radiotherapy, or radical prostatectomy. According to the NCCN guidelines, there is no recommendation for any specific regimen over others, but in clinical practice, the judgement usually depends on the tumor volume, as applied from a subgroup analysis of the original RCTs [35]. A range of RCTs were conducted to investigate the efficacy of each combination treatment regimen, and some of them were reported repeatedly with a longer follow-up using the same cohort. In addition, there were many SRMAs that tried to pool the data from RCTs which compared each intervention. Among these SRMAs, most studies demonstrated the same direction of outcome, whereas some were still conflicting. This study summarized and displayed the reported outcomes from the previous SRMAs, not only from direct comparisons but also indirect comparisons. This study’s results were analyzed using umbrella review methods, which showed a high certainty of evidence that the combination treatments were more beneficial than ADT alone in both OS and PFS outcomes. On the other hand, there was no OS advantage of any combination regimen over the others. Some studies assessed comparisons between docetaxel with ADT and ADT alone, and a few recent studies followed the patients with longer follow-up times and found slightly different HRs for OS. For the PFS of combination treatments, most of the results were still conflicting. In addition, the comparisons among combination regimens are based solely on indirect comparisons from network meta-analyses. As a result, further RCTs among combination treatments may be necessary to affirm clear results. Regarding the AEs, cardiovascular risks tend to decrease more in patients receiving GnRH antagonists than GnRH agonists, although this was not significant [35]. However, life-threatening AEs were not significantly different between abiraterone, enzalutamide, and docetaxel compared with ADT alone.

To further evaluate the possibility of why the SRMAs showed the same direction, one of the strategies is to assess the degree of overlap. For the overall included SRMAs, the percentage of overlap was 21.3% with a CA of 7% and CCA of 3.8%, implying a low degree of overlap. However, when focusing on only the OS outcome, the overlap was very high, with a 51.5% overlap, CA of 21%, and CCA of 16.4%, suggesting that the RCTs included in each SRMAs were similar and correlating with the fact that each intervention had unique and large scale RCTs that were referred as the prototype of those comparisons. The concordant results from each SRMA should be interpreted with caution. The possible explanation for the overall lower degree of overlap was probably from including AE outcomes in this study’s review. The excessive significance test, which aimed to evaluate if the observed number of studies with statistically significant findings was more than what would be expected by chance, found no excessive significance in all SRMAs, indicating no reporting biases.

This study has several strengths. This study performed a systematic review and logically reported the outcome from each SRMA. In addition, to our knowledge, this is the first umbrella review reporting metastatic HSPC treatment. There were limitations of this study, such as this study did not pool the effect size of each outcome and so it was not possible to accurately compare the outcomes among the treatments. Further well-designed studies are suggested to obtain further conclusions.

## 5. Conclusions

There are many treatment options for metastatic HSPS patients, either ADT alone or combination treatments. Regarding the previous SRMAs, combination treatments demonstrated clear benefits in OS and PFS over ADT alone with more AEs. Further studies are needed to compare combination treatments.

## Figures and Tables

**Figure 1 cancers-15-05714-f001:**
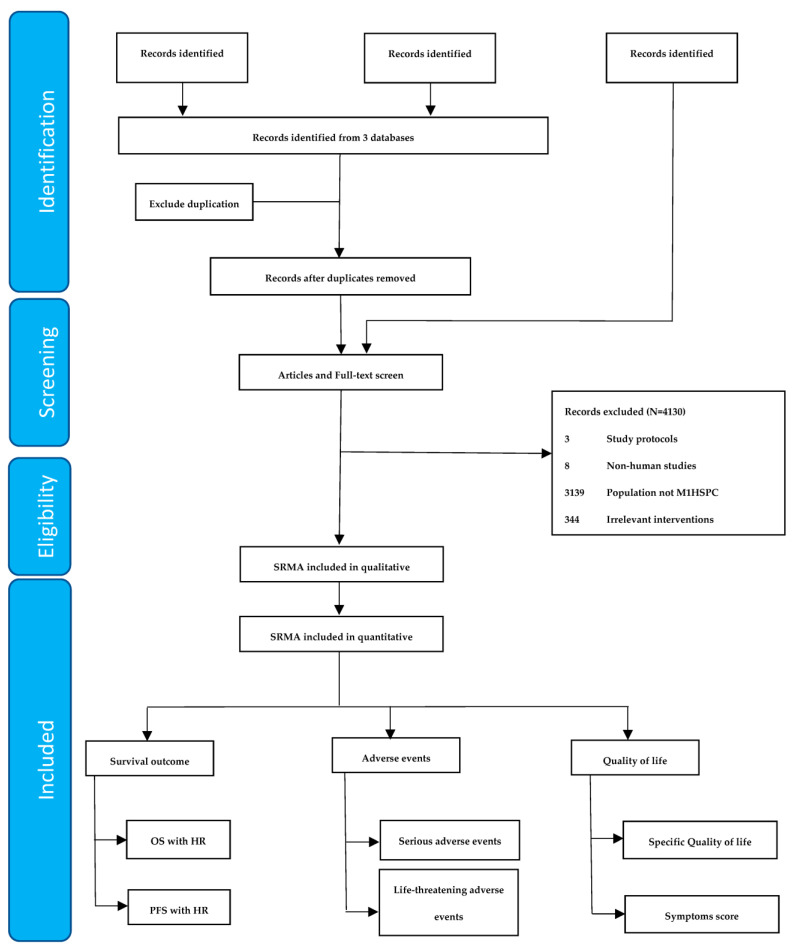
Flow diagram showing the selection of articles for review.

**Figure 2 cancers-15-05714-f002:**
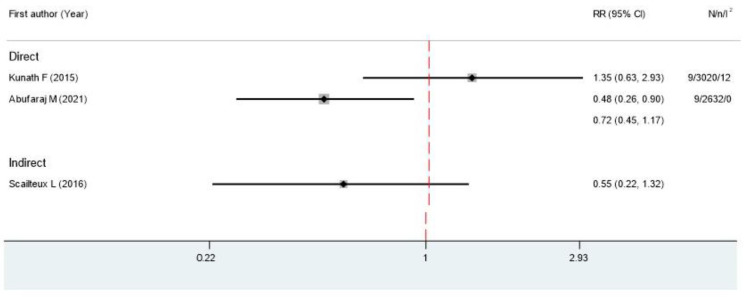
Forest plots show the risk ratio of the mortality rate for metastatic HSPC patients between GnRH antagonists and GnRH agonists; RR, relative risks; CI, confidence intervals; N, number of primary studies; n, total number of patients; I^2^, degrees of heterogeneity from Higgin’s I^2^ statistics [12,13,16].

**Figure 3 cancers-15-05714-f003:**
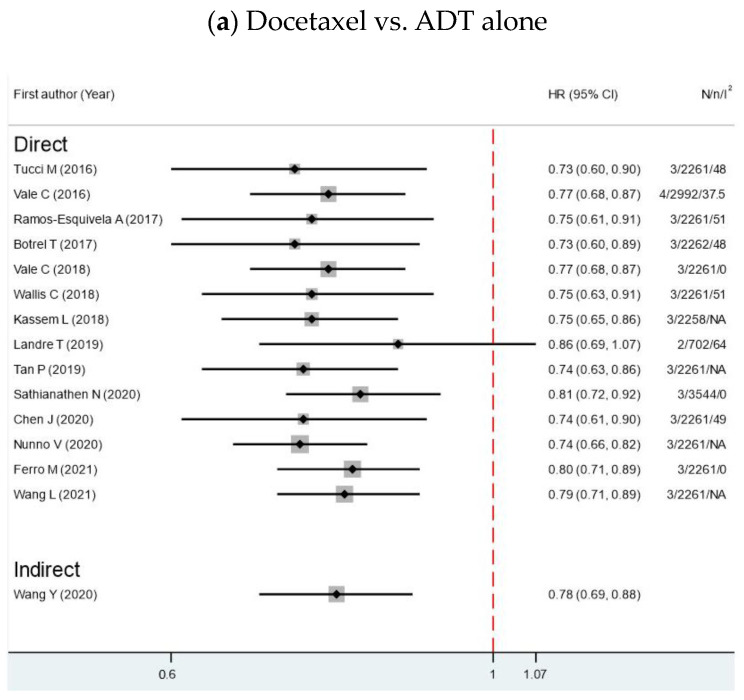
Forest plots show hazard ratios of OS for metastatic HSPC patients between 2 interventions. HR, hazard ratios; CI, confidence intervals; N, number of primary studies; n, total number of patients; I^2^, degrees of heterogeneity from Higgin’s I^2^ statistics [8,17,18,19,20,21,22,23,24,25,26,27,28,29,30,31,32,33].

**Figure 4 cancers-15-05714-f004:**
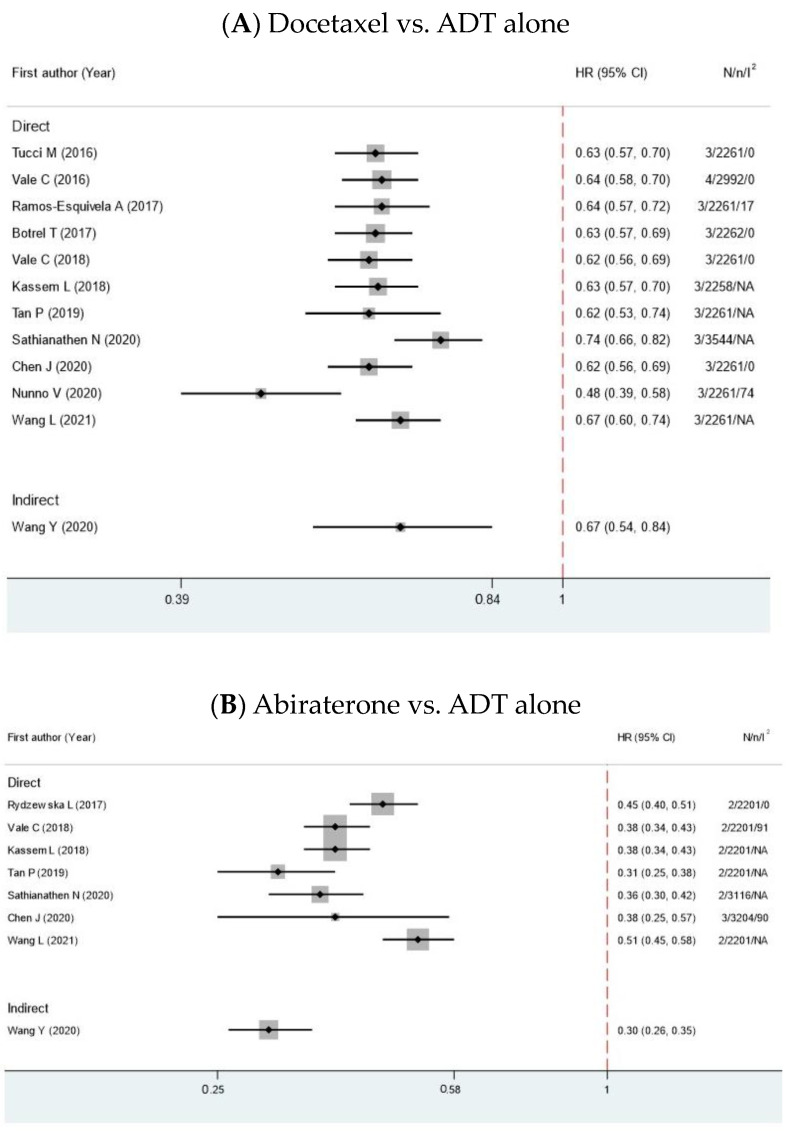
Forest plots show hazard ratios of PFS for metastatic HSPC patients between 2 interventions. HR, hazard ratios; CI, confidence intervals; N, number of primary studies; n, total number of patients; I^2^, degrees of heterogeneity from Higgin’s I^2^ statistics [6,7,17,18,19,20,21,23,24,25,26,27,29,30,32,33].

**Figure 5 cancers-15-05714-f005:**
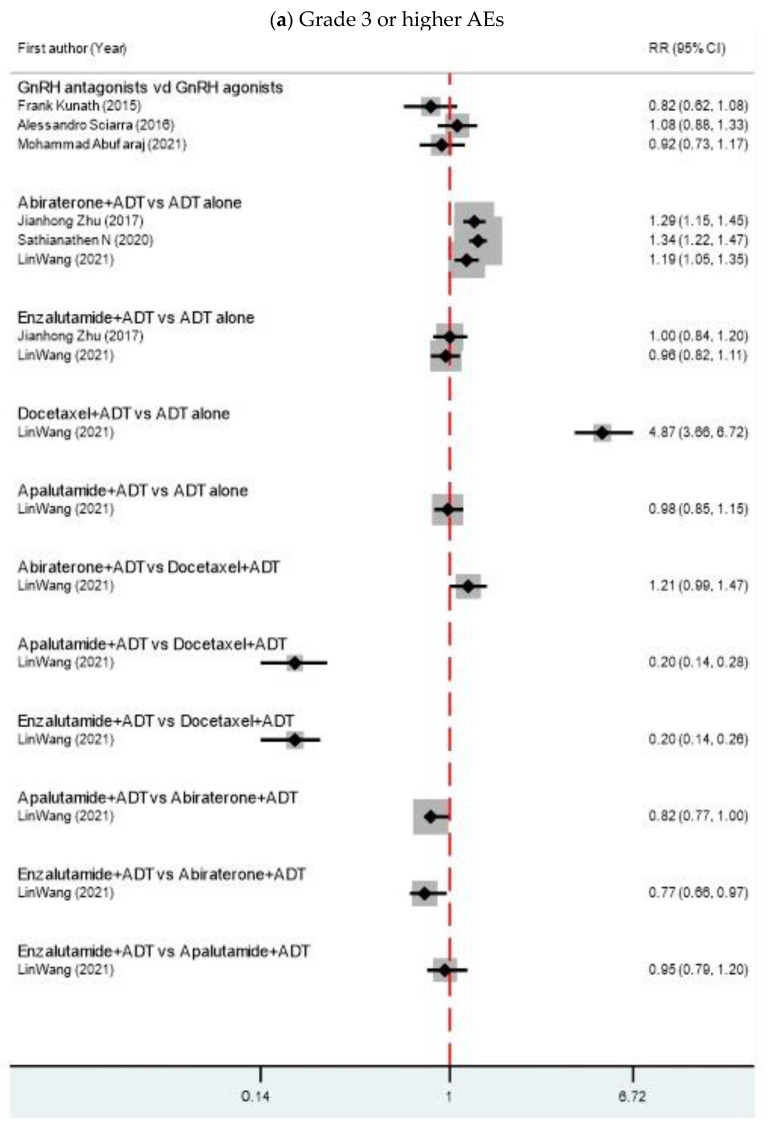
Forest plots show risk ratios of AEs in metastatic HSPC patients between 2 interventions. RR, relative risks; CI, confidence intervals [6,13,14,15,16,26,29,32,34].

**Table 1 cancers-15-05714-t001:** Study characteristics of the included systematic reviews and meta-analyses.

Author	Year	MA Type	Comparison	Search Time	N	n	Years RCT Published	Interested Outcome	Median Age (IQR)	Median PSA (IQR)
Ferro M	2021	NMA	ADT vs. Docetaxel	30 October 2020	7	2261	2016–2020	OS	66.9 (36.0–88.0)	50.9 (0.2–8540.1)
Wang L	2021	NMA	ADT vs. Docetaxel vs. Abiraterone vs. Apalutamide vs. Enzalutamide	5 November 2019	19	7287	2013–2019	OS, PFS, AE	N/A	N/A
Abufaruj M	2021	DMA	GnRH agonists vs. GnRH antagonists	April 2020	8	2632	2008–2019	AE, PSA-progression, overall mortality	N/A	N/A
Sathianathen N	2020	NMA	ADT vs. Docetaxel vs. Abiraterone vs. Apalutamide vs. Enzalutamide	5 November 2019	10	7287	2013–2019	OS, PFS, AE	67.0 (33.0–92.0)	51.0 (Range 0.2–8450.1)
Chen J	2020	NMA	ADT vs. Docetaxel vs. Abiraterone vs. Apalutamide vs. Enzalutamide	25 September 2019	11	11,174	2013–2019	OS, PFS	67.0	51.0
Wang Y	2020	NMA	ADT vs. Docetaxel vs. Abiraterone vs. Apalutamide vs. Enzalutamide *	7 May 2020	8	8701	2015–2019	OS, PFS	N/A	N/A
Nunno V	2020	DMA	ADT vs. Docetaxel vs. Abiraterone *	15 September 2019	10	8324	2013–2019	OS, PFS, AE	N/A	N/A
Sathianathen N	2020	DMA	ADT vs. Abiraterone	15 May 2020	9	2201	2013–2019	OS, AE, FACT-P	67.0 (33.0–92.0)	53.5 (19.0–165.0)
Landre T	2019	DMA	ADT vs. Docetaxel, ADT vs. Abiraterone	Jan 2018	4	2264	2015–2017	OS, PFS	N/A	N/A
Vale C	2018	NMA	ADT vs. Docetaxel vs. Abiraterone	30 September 30th	5	6204	2016–2017	OS, PFS	N/A	N/A
Feyerabend S	2018	NMA	ADT vs. Docetaxel vs. Abiraterone	July 2017	10	4804	2015–2018	OS, PFS, FACT-P	N/A	N/A
Wallis C	2018	NMA	ADT vs. Docetaxel vs. Abiraterone	4 August 2017	5	6067	2015–2017	OS, AE	65.0 (33.0–92.0)	51.0 (5.0–165.0)
Kassem L	2018	NMA	ADT vs. Docetaxel vs. Abiraterone	June 2017	7	7469	2013–2017	OS, PFS, AE	64.5	N/A
Tan P	2018	NMA	ADT vs. Docetaxel vs. Abiraterone *	26 August 2017	6	3877	2013–2017	OS, PFS, AE	N/A	N/A
Aoun F	2017	NMA	ADT vs. Docetaxel vs. Abiraterone	not mentioned	5	4827	2015–2017	OS, PFS, AE	N/A	N/A
Rydzewska L	2017	DMA	ADT vs. Abiraterone	May 2017	2	2201	2017	OS, PFS, AE	67.0 (33.0–92.0)	N/A
Ramos-Esquivela A	2016	DMA	ADT vs. Docetaxel *	1 October 2015	3	2261	2013–2016	OS, PFS, AE	N/A	53.5 (5.0–181.0)
Botrel T	2016	DMA	ADT vs. Docetaxel	not mentioned	7	2264	2013–2016	OS, PFS, AE	63.0	38.0
Hosseini S	2016	DMA	GnRH agonists vs. GnRH antagonists	up to 2014	6	2296	2008–2013	AE	72.0 (50.0–98.0)	19.8 (8.2–68.0)
Lei J	2016	DMA	ADT vs. Docetaxel *	August 2014	2	1175	2013–2014	OS in OR	N/A	N/A
Tucci M	2016	DMA	ADT vs. Docetaxel	August 2015	4	2951	2013–2015	OS, PFS	64.0 (39.0–91.0)	39.0 (5.0–127.0)
Vale C	2016	DMA	ADT vs. Docetaxel	30 September 2015	4	2992	2013–2016	OS, PFS	64.0 (36.0–91.0)	N/A
Sciarra A	2016	DMA	GnRH agonists vs. GnRH antagonists	30 July 2015	7	1719	2008–2015	AE	71.9 (51.0–98.0)	19.1 (0.01–12,861.0)
Kunath F	2015	DMA	GnRH agonists vs. GnRH antagonists	March 2015	17 **	3641 **	2001–2014	AE	N/A	N/A
Zhu X	2019	DMA	ADT vs. Enzalutamide	July 2019	7	7347	2012–2019	AE	N/A	N/A
Zhu J	2017	DMA	ADT vs. Abiraterone vs. Enzalutamide	June 2017	10	9520	2011–2017	AE	N/A	N/A
Scailteux L	2016	NMA	Bilateral orchiectomy vs. GnRH agonists vs. GnRH antagonists	July 2014	57 **	31,037 **	1985–2013	AE	N/A	N/A

* included only the interesting arm. ** included only RCTs. Abbreviations: ADT, androgen deprivation therapy; AE, adverse events; DMA, direct meta-analysis; FACT-P, The Functional Assessment of Cancer Therapy-Prostate; GnRH, gonadotropin hormone-releasing hormone; IQR, interquartile range; N, total number of included RCTs; n, number of patients in the SRMA; NMA, network meta-analysis; OR, odds ratio; OS, overall survival; PFS, progression-free survival; PSA, prostate-specific antigen; RCT, randomized controlled trial. 3.1. Mortality rate.

**Table 2 cancers-15-05714-t002:** Risk of bias assessment using ROBIS.

Author	Year	1. Study Eligibility Criteria	2. Identification and Selection of Studies	3. Data Collection and Study Appraisal	4. Synthesis and Findings	5. Risk of Bias in the Review
1. Ferro M	2021	☹	☺	☹	☹	☹
2. Wang L	2021	☺	☺	☺	☺	☺
3. Mohammad A	2021	☹	☺	☺	☺	☺
4. Sathianathen N	2020	☺	☺	☺	☺	☺
5. Chen J	2020	☺	☺	☺	☺	☺
6. Wang Y	2020	☺	☺	☺	☺	☺
7. Nunno VD	2020	☺	☺	☺	☺	☺
8. Sathianathen N	2020	☺	☺	☺	☺	☺
9. Landre T	2019	☺	☹	☹	☺	☹
10. Vale CL	2018	☺	☺	☺	☺	☺
11. Feyerabend S	2018	☺	☺	☺	☹	☺
12. Wallis CJD	2018	☺	☺	☺	☺	☺
13. Kassem L	2018	☺	☺	☹	☹	☺
14. Tan PS	2018	☹	☹	☹	☹	☹
15. Aoun F	2017	☹	☹	☹	☹	☹
16. Rydzewska LHM	2017	☺	☺	☺	☺	☺
17. Ramos-Esquivela A	2016	☺	☺	☺	☺	☺
18. Botrel TEA	2016	☹	☺	☺	☺	☺
19. Hosseini SA	2016	☹	☺	☺	☹	☹
20. Lei J	2016	☺	☺	☺	☺	☺
21. Tucci M	2016	☺	☺	☹	☺	☹
22. Vale CL	2016	☺	☺	☺	☺	☺
23. Sciarra A	2016	☺	☺	☺	☺	☺
24. Kunath F	2015	☺	☺	☺	☺	☺
25. Zhu X	2019	☺	☺	☺	☺	☺
26. Zhu J	2017	☺	☺	☺	☺	☺
27. Scailteux L	2016	☺	☺	☺	☺	☺

☺, Low risk of bias; ☹, High risk of bias

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
