# Peer review of "Efficacy of Treatment for Metastatic Hormone-Sensitive Prostate Cancer: An Umbrella Review of Systematic Reviews and Meta-Analyses"

_cancers, 2023, doi:10.3390/cancers15245714_

Round 1
Reviewer 1 Report
Comments and Suggestions for Authors
The study has sufficient merit to be considered for publication, although major revisions are required.
1. Methods and methodology are robust.
2. Results, conclusions, and limitations are well presented.
3. The bibliography contains the most up-to-date references.
4. Tables and graphics are clearly described.
5. The authors should give more detailed information on the side effects and contraindications (https://doi.org/10.3389%2Ffendo.2021.695170) , predictor of efficacy (https://doi.org/10.1016/j.critrevonc.2020.102992) and patients’ comorbidities (https://doi.org/10.1159%2F000509434) that could influence the treatment of the antiandrogenic therapy. A lecture on these interesting studies could enhance the scientific value of the paper.
Comments on the Quality of English Language
Minor editing
Author Response
Reviewer1 comments
- The authors should give more detailed information on the side effects and contraindications (https://doi.org/10.3389%2Ffendo.2021.695170) , predictor of efficacy (https://doi.org/10.1016/j.critrevonc.2020.102992) and patients’ comorbidities (https://doi.org/10.1159%2F000509434) that could influence the treatment of the antiandrogenic therapy. A lecture on these interesting studies could enhance the scientific value of the paper.
Response to reviewer: We have added more details about adverse events of antiandrogenic treatments along with citations in the discussion, see page 19 line 22. However, we do not include information of predictors because this is out of our scope given the focus on treatment efficacy, not the effects of predictors. We hope that you do not mind for this.

Reviewer 2 Report
Comments and Suggestions for Authors
1. Abstract: Some senteces are not clear for me. This is an example. “We systematically appraised the results, overlapping, excessive significant test and the quality of the studies.”
2. “ADT, GnRH” .The acronyms should be spelled out.
3. Introduction:
3.1 There are two main issues in the following sentence that may be misleading.
“Meanwhile, prostate cancer screening programme using PSA is more popular and recommended in subgroup of population in this period. Strong evidence indicate that prostate cancer screening reduces the rates of advanced disease at the time of diagnosis by 75%”
First: the concept of PSA as screening test. Recently, it has been discussed about the relevance of applying PSA testing to men at risk, to improve the risk-benefit ratio of measuring PSA. The recommendations in the Guidelines have been changed in the course of 2018 since the high rate of false positive tests evidenced by the literature. You can find all these concepts in this paper: Ferraro S, et al. Serum prostate specific antigen (PSA) testing for early detection of prostate cancer: Managing the gap between clinical and laboratory practice. Clin Chem 2021;67:602-609
Second: Consequently, “the reduction of 75% of tumors” reported by the authors is under debate, since it has been reported an overtreatment. The greatest part of Prostate Cancer tumors is of low risk and patients do not benefit from treatment. For this reason, prediction algorithms including PSA values are used to identify those tumors requiring to be treated. Ferraro S, et al. Individual Risk Prediction of Advanced Prostate Cancer based on the combination between total Prostate-Specific Antigen (PSA) and free to total PSA ratioClin Chem Lab Med. 2023
3.2 SRMA should be spelled out the first time in the introduction.
3.3 Table 1. Total N of SRMA: You should confirm or is this the total number of patients?
3.4 The median and percentiles of age of the patients has to be reported since age is relevant for the prognostic outcome.
3.5 PSA has been reported in the introduction as diagnostic tool. However PSA is monitored also in this case to evaluate the biochemical response to the treatment, and it is a relevant prognostic factor. If it is available for all the studies it is relevant to report it in the table 1. In particular median and percentiles values of PSA should be considered.
3.6 “ patients more than 70 years of age”. “Older than?” Extensive English editing is required.
Comments on the Quality of English Language
Extensive editing is required
Author Response
Reviewer2 comments
- Abstract: Some sentences are not clear for me.This is an example. “We systematically appraised the results, overlapping, excessive significant test and the quality of the studies.”
Response to reviewer: We have re-written these sentences in the abstract (page1 lines 32-34).
- “ADT, GnRH”. The acronyms should be spelled out.
Response to reviewer: We have defined these abbreviations in page1 lines 17, 20 and page 2 lines 61, 80.
- Introduction:
3.1 There are two main issues in the following sentence that may be misleading.
“Meanwhile, prostate cancer screening programme using PSA is more popular and recommended in subgroup of population in this period. Strong evidence indicate that prostate cancer screening reduces the rates of advanced disease at the time of diagnosis by 75%”
First: the concept of PSA as screening test. Recently, it has been discussed about the relevance of applying PSA testing to men at risk, to improve the risk-benefit ratio of measuring PSA. The recommendations in the Guidelines have been changed in the course of 2018 since the high rate of false positive tests evidenced by the literature. You can find all these concepts in this paper: Ferraro S, et al. Serum prostate specific antigen (PSA) testing for early detection of prostate cancer: Managing the gap between clinical and laboratory practice. Clin Chem 2021;67:602-609
Second: Consequently, “the reduction of 75% of tumors” reported by the authors is under debate, since it has been reported an overtreatment. The greatest part of Prostate Cancer tumors is of low risk and patients do not benefit from treatment. For this reason, prediction algorithms including PSA values are used to identify those tumors requiring to be treated. Ferraro S, et al. Individual Risk Prediction of Advanced Prostate Cancer based on the combination between total Prostate-Specific Antigen (PSA) and free to total PSA ratio Clin Chem Lab Med. 2023
Response to reviewer: We have re-written about PSA screening recommendation based on the American Urological Association guideline 2023 as in page 2 line 55, and also remove the point that still under the debate.
3.2 SRMA should be spelled out the first time in the introduction.
Response to reviewer: Done, see page 2 line 71.
3.3 Table 1. Total N of SRMA: You should confirm or is this the total number of patients?
Response to reviewer: We have described abbreviations of N, n, and others in the footnote of Table 1.
3.4 The median and percentiles of age of the patients has to be reported since age is relevant for the prognostic outcome.
Response to reviewer: We have added the median age (IQR)for each SRMA in Table1.
3.5 PSA has been reported in the introduction as a diagnostic tool. However, PSA is monitored also in this case to evaluate the biochemical response to the treatment, and it is a relevant prognostic factor. If it is available for all the studies it is relevant to report it in the table 1. In particular median and percentiles values of PSA should be considered.
Response to reviewer: Done, see Table1.
3.6 “patients more than 70 years of age”. “Older than?” Extensive English editing is required.
Response to reviewer: We have sent the manuscript for English editing by Mr. Stephen Pinder who is a native speaker specializing in medical English.

Round 2
Reviewer 1 Report
Comments and Suggestions for Authors
Authors answered all comments and suggestions.
Comments on the Quality of English Language
Minor editing
Author Response
Thank you.

Reviewer 2 Report
Comments and Suggestions for Authors
The sentence I have previously highlighed has been changed, "lines 55-57:..." clinicians should provide information for those who would benefit from prostate cancer screening and making a decision based on a person’s values and preferences "quote only the reference to the guideline 2023 which now is lacking.
Furthermore, at the end of this sentence, I reccommend to add a sentence to higlight the improved role of PSA in supporting clinicians, and the recent improvement of interpretative criteria. "Prediction algorithms including PSA values are used to support urologists to identify those tumors requiring to be treated." you can find this relevant concept at this reference: Ferraro S, et al. Individual Risk Prediction of Advanced Prostate Cancer based on the combination between total Prostate-Specific Antigen (PSA) and free to total PSA ratio Clin Chem Lab Med. 2023
This is relevant since then you introduce advanced cancer, and you report in the table PSA values. Curiously, these latter quite overlap accross most studies on ADT.
Author Response
Reviewer2 comments
- The sentence I have previously highlighed has been changed, "lines 55-57:..." clinicians should provide information for those who would benefit from prostate cancer screening and making a decision based on a person’s values and preferences "quote only the reference to the guideline 2023 which now is lacking.
Furthermore, at the end of this sentence, I reccommend to add a sentence to higlight the improved role of PSA in supporting clinicians, and the recent improvement of interpretative criteria. "Prediction algorithms including PSA values are used to support urologists to identify those tumors requiring to be treated." you can find this relevant concept at this reference: Ferraro S, et al. Individual Risk Prediction of Advanced Prostate Cancer based on the combination between total Prostate-Specific Antigen (PSA) and free to total PSA ratio Clin Chem Lab Med. 2023
Response to reviewer: We have added the sentences mentioned about prediction nomogram into introduction part, page2 line 58-60.
- This is relevant since then you introduce advanced cancer, and you report in the table PSA values. Curiously, these latter quite overlap across most studies on ADT.
Response to reviewer: Yes. As we reported the degree of overlapping of overall survival outcome is very high(16.4%), implying that the same RCTs are included in each meta-analysis.
